# SARS-CoV-2 Rapid Antigen Testing of Symptomatic and Asymptomatic Individuals on the University of Arizona Campus

**DOI:** 10.3390/biomedicines9050539

**Published:** 2021-05-12

**Authors:** David T. Harris, Michael Badowski, Brandon Jernigan, Ryan Sprissler, Taylor Edwards, Randall Cohen, Stephen Paul, Nirav Merchant, Craig C. Weinkauf, Christian Bime, Heidi E. Erickson, Billie Bixby, Sairam Parthasarathy, Sachin Chaudhary, Bhupinder Natt, Elaine Cristan, Tammer El Aini, Franz Rischard, Janet Campion, Madhav Chopra, Michael Insel, Afshin Sam, James L. Knepler, Kenneth Knox, Jarrod Mosier, Catherine Spier, Michael D. Dake

**Affiliations:** 1Biorepository, College of Medicine-Tucson, University of Arizona Health Sciences, University of Arizona, Tucson, AZ 85724, USA; badowski@arizona.edu (M.B.); brandonjernigan@arizona.edu (B.J.); 2Department of Immunobiology & Medicine, College of Medicine-Tucson, University of Arizona Health Sciences, University of Arizona, Tucson, AZ 85724, USA; 3College of Medicine-Tucson, University of Arizona Genetics Core, University of Arizona, Tucson, AZ 85724, USA; ryans1@ariziona.edu (R.S.); taylore@arizona.edu (T.E.); 4Department of Athletic, University of Arizona, Tucson, AZ 85724, USA; rpcohen@arizona.edu (R.C.); spaul@arizona.edu (S.P.); 5Data Science Institute, University of Arizona, Tucson, AZ 85724, USA; nirav@arizona.edu; 6Department of Surgery, College of Medicine-Tucson, University of Arizona, Tucson, AZ 85724, USA; ccweinkauf@arizona.edu; 7Division of Pulmonary, Allergy, Critical Care and Sleep Medicine, College of Medicine-Tucson, University of Arizona, Tucson, AZ 85724, USA; cbime@arizona.edu (C.B.); herickso@arizona.edu (H.E.E.); bixbyba@arizona.edu (B.B.); sparthasarathy@deptofmed.arizona.edu (S.P.); sachin@deptofmed.arizona.edu (S.C.); bnatt@deptofmed.arizona.edu (B.N.); cristan@deptofmed.arizona.edu (E.C.); telaini@deptofmed.arizona.edu (T.E.A.); rischard@arizona.edu (F.R.); jcampion@deptofmed.arizona.edu (J.C.); chopramv@deptofmed.arizona.edu (M.C.); minsel@deptofmed.arizona.edu (M.I.); asam@deptofmed.arizona.edu (A.S.); jknepler@arizona.edu (J.L.K.); 8Department of Medicine, University of Arizona-Phoenix, Phoenix, AZ 85724, USA; kknox@arizona.edu; 9Department of Emergency Medicine, College of Medicine-Tucson, Tucson, AZ 85724, USA; jmosier@arizona.edu; 10Department of Pathology, University of Arizona College of Medicine-Tucson, University of Arizona, Tucson, AZ 85724, USA; catherine.spier@bannerhealth.com; 11Office of the Senior Vice-President for Health Sciences, University of Arizona, Tucson, AZ 85724, USA; mddake@arizona.edu

**Keywords:** rapid antigen test, PCR, SARS-CoV-2, asymptomatic, diagnostic screening

## Abstract

SARS-CoV-2, the cause of COVID19, has caused a pandemic that has infected more than 80 M and killed more than 1.6 M persons worldwide. In the US as of December 2020, it has infected more than 32 M people while causing more than 570,000 deaths. As the pandemic persists, there has been a public demand to reopen schools and university campuses. To consider these demands, it is necessary to rapidly identify those individuals infected with the virus and isolate them so that disease transmission can be stopped. In the present study, we examined the sensitivity of the Quidel Rapid Antigen test for use in screening both symptomatic and asymptomatic individuals at the University of Arizona from June to August 2020. A total of 885 symptomatic and 1551 asymptomatic subjects were assessed by antigen testing and real-time PCR testing. The sensitivity of the test for both symptomatic and asymptomatic persons was between 82 and 90%, with some caveats.

## 1. Introduction

In less than one year, the novel coronavirus, SARS-CoV-2, the causative agent of COVID-19, has jumped from animals to humans, causing a pandemic that has infected more than 80 M and killed more than 12 M persons worldwide. In the US alone, as of February 2021, it has infected more than 32 M people while causing approximately 570,000 deaths [1]. Both sets of numbers continue to mount daily, with fears of a surge of infections ever-present. As the pandemic persists, there has been great public demand to “reopen” cities, workspaces and places of social interaction such as schools and university campuses despite the risks involved. To consider these demands it is necessary to rapidly identify the virus-infected individuals and isolate them so that disease transmission can be stopped.

Tracking and tracing efforts that are required to establish the identity of secondary infections depend on the rapid identification of the primary infected person to limit viral spread. The source of infection must be isolated and quarantined so they may be observed and treated as necessary. Viral testing strategies must be specific for the disease, sensitive to low (possibly asymptomatic) viral loads and return results to those responsible in as short a time as possible [2]. Typically, polymerase chain reaction (PCR) assays have been the gold standard for viral diagnostic testing [3,4], but other testing options are available. Unfortunately, the PCR test is not generally a point-of-care test and often takes 48–72 h to return results, is relatively expensive at scale, has been plagued by reagent scarcity and may even be too sensitive at times [2,3,4].

There are other testing options, however, that may be sufficiently sensitive and disease-specific to achieve the rapid identification of infected patients and facilitate early isolation and contact tracing. Such tests may be able to compensate for reduced sensitivity as compared to PCR testing by being amenable to multiple rounds of testing combined with the rapid return of results within minutes to hours. As long as these alternative testing approaches are safe, easy to perform and inexpensive, there could be significant advantages. If these tests are capable of being self-administered and include the return of results documentation, then such tests might become a preferred choice when screening large populations of individuals in a high throughput manner.

Such a testing strategy became commercially available from Quidel, Inc. (San Diego, CA, USA) to analyze subjects for the presence of the SARS-CoV-2 viral antigen Ag) [5]. The assay (Sofia SARS Antigen FIA) detects the presence of antigen produced by a live virus. The test reports the result within 15–30 min, can be scaled to perform thousands of tests per day, has US Food and Drug Administration (FDA) Emergency Use Authorization (EUA) for symptomatic subjects, is relatively inexpensive at USD 23/test, and can be performed using self-administered (anterior) nasal swabs. Quidel is a company with a history of producing FDA-approved rapid diagnostic tests (such as for influenza), lending confidence to purported claims. The testing device, the Sofia 2, has an internal storage device, along with a flash/USB reporting option, and data can be directly exported to the cloud via 3G connections for inclusion in subject medical records. Quidel has reported the sensitivity of its SARS-CoV-2 rapid Ag test to be 93% with near 100% specificity [5], (**FDA EUA Application**) but these results were not based on testing large numbers of “real-world” samples.

In the present study, we examined the sensitivity of the Quidel Rapid Ag test for use in screening both symptomatic and asymptomatic individuals in the student and staff community at the University of Arizona from June to August 2020. After test validation, a total of 885 symptomatic and 1551 asymptomatic subjects were assessed by both antigen testing and real-time PCR (RT-PCR or PCR) testing performed side-by-side at the same time on the same patient.

## 2. Materials and Methods

### 2.1. Subjects

All human subjects in the study were students, faculty, or staff at the University of Arizona. Participation was voluntary. Symptomatic subjects were tested within 5 days of symptom onset. Asymptomatic subjects were asked to volunteer for testing as a screen for return to campus activities, and none had symptoms of infection or had been exposed to anyone with symptoms. All testing was performed by CLIA and/or CAP-accredited laboratories under the standard of care. An institutional review determined that the study did not require consent as long as all data was de-identified upon receipt and that all data pertaining to this study were de-identified prior to publication. All collections were performed between June and August 2020. In addition, a small number of COVID-19-positive patients from the local university ICU provided samples as part of the initial validation studies of the assay (with consent).

### 2.2. Biospecimen Collection

Rapid antigen (Ag) testing specimens were collected using foam swabs provided by the manufacturer. Dry anterior nasal collections were self-performed for both nostrils under supervision. Dry swabs were placed in a barcoded 15 mL conical tube, kept in an insulated cooler to maintain 65–80 °F (18–27 °C) and transported within 60 min of collection to the laboratory. PCR specimens were collected from one nostril using CDC-approved nasopharyngeal (NP) swabs (Miraclean #MRC 96000, Tongle, Longgang, Shenzhen, China) with medical assistance. The swab was then placed in a VMT transport buffer and brought to the laboratory for testing within 60 min. NP and Ag nasal specimens were collected in that order at the same time from individuals in this study. Tested subjects were classified as being antigen-positive/PCR-positive, antigen-negative/PCR-negative, antigen-positive/PCR-negative or antigen-negative/PCR-positive based on the combination of test results.

### 2.3. PCR Testing

PCR testing utilized a CDC-approved primer set with a detected/not detected cut-off of 40 cycles (CDC 2019-Novel Coronavirus (2019-nCoV) Real-Time RT-PCR Diagnostic Panel (N1, N2 and RP primer set) ordered through Integrated DNA Technologies, Inc. (IDT): Catalog #2019-nCoVEUA-01 Diagnostic Panel Box #1; (https://www.cdc.gov/coronavirus/2019-ncov/lab/virus-requests.html) (accessed on 1 February 2021) [6,7]. Tests were performed according to the manufacturer’s instructions.

### 2.4. Rapid Antigen Testing

The Quidel Sofia 2 SARS antigen FIA test displayed a digital readout as positive or negative based on the ratio of relative fluorescence units to background in an LFT (lateral flow test) assay termed the S/CO (signal versus control) ratio [5]. Ratios above 1.0 are considered positive results, and ratios can range from 1–600 depending on the level of viral infection (viral load). Samples were processed according to the manufacturer’s instruction within 90 min of arrival at the laboratory and analyzed using the Sofia 2 instrument.

## 3. Results

The sensitivity and specificity of the rapid Ag assay were initially assessed by performing rapid antigen testing using three types of biospecimens: “spiked” positive and negative samples provided by Quidel, Inc.; samples obtained from suspected, symptomatic, critically ill patients admitted to intensive care units; and samples obtained from asymptomatic campus students and staff (Table 1). Anterior nasal samples were collected with dry swabs, processed and tested as per the manufacturer’s instructions. PCR testing specimens were simultaneously collected using NP swabs, placed into a viral transport media (VTM), processed and analyzed within 12 h. Concordant results were observed for rapid antigen testing and PCR testing for the majority of samples examined (32 of 35 samples), whether spiked samples, suspected COVID-19-positive samples or asymptomatic patients were analyzed. Discordant findings where subjects were antigen-negative but PCR-positive for the presence of the virus were observed for specimens with high PCR cycle thresholds (of 31 and above). The overall results obtained in these studies provided a determination of 91.4% sensitivity with 100% specificity for the antigen test, similar to that claimed by the vendor.

A select group of initially asymptomatic patients from the validation studies who tested positive during the validation studies were followed and tested for an additional 1–2 weeks using the rapid antigen test and PCR test as before. As shown in Table 2, during subsequent testing once PCR cycle thresholds (ct) for locus N1 amplification were higher than 31 cycles, the rapid antigen test was unable to detect the presence of the virus. Preliminary results from such subjects have indicated that anti-virus antibody titers were present during those later test times when subjects displayed high ct values and that patients may not be contagious due to low viral loads or the absence of live virus. Although preliminary, these results may also suggest that the shedding of live virus as indicated by a positive rapid antigen test may only persist for a week (or so), despite continued reactivity by viral PCR analysis.

To explore the limits of the rapid antigen assay, 885 students and staff presenting at university campus health were tested, including both those exposed to the virus (i.e., in close contact with someone that tested positive for the virus) and those symptomatic with cough, fever, aches, etc. were tested. As shown in Table 3, more than half of all subjects (64.8%) presenting with symptomatic complaints were negative for the virus by both the PCR and rapid antigen tests. A total of 305 individuals tested positive by PCR (34.5%) while a total of 258 persons tested positive by the rapid antigen test (29.2%). Discordant PCR and rapid antigen testing results were once again observed in biospecimens with high PCR cycle thresholds (average ct of 34). It appeared that the Quidel rapid antigen test did not detect approximately 6% of the positive results detected by the PCR test. These samples displayed cts with 8 cycles higher on average. This finding implies that the COVID-19-positive samples that the Quidel test “missed” contained on average 256 times less viral load than the positive samples it could detect. Calculations from these results indicated a sensitivity of 82.3%, while the specificity of the rapid antigen assay was 98.2% based on a finding of 7 antigen-positive, PCR-negative results out of 885 total results. The positive predictive value (PPV) and negative predictive value (NPV) were calculated to be 97.3% and 91.4%, respectively. A similar assessment to that described was performed with a large sampling of the university campus staff and student body as shown in Table 3. All subjects were randomly selected and asymptomatic at the time of both PCR and rapid antigen testing. The vast majority of subjects were virus-negative by both the PCR and the rapid antigen test (98%), as might be expected.

Based on these findings, further analyses were performed to examine the role of PCR cycle time (i.e., potential viral load) in the detection of the virus by the rapid antigen test. It was observed that symptomatic patients displayed lower cycle thresholds in the PCR analysis (see Figure 1) regardless of whether they were also Ag-positive for the virus. Reasons for this observation are not straightforward, but it may be that some SARS-CoV-2 infections are missed due to sampling or that lower PCR cycle thresholds may be a result of an unrecognized inflammatory immune reaction. Regardless, symptomatic patients seemed to require fewer PCR cycle times to detect the virus. Violin dot plots of the findings (Figure 1) demonstrated that subjects that were PCR-positive but Ag-negative displayed cycle times clustering at the higher end of the scale.

The results observed in Figure 1 led us to further examine the sensitivity of the rapid Ag test in asymptomatic subjects. Table 3 dissects the effects of PCR cycle threshold on test concordance in more detail. In each of the 4 groups, the subjects with ct > 30 were segregated from the other subjects in each of the populations. As shown, this stratification resulted in 243 out of 277 (88%) symptomatic subjects and 6 out of 7 (86%) asymptomatic subjects with cycle times below 30 being both Ag and PCR test concordant. These analyses confirmed the conclusions implied in Figure 1 demonstrating that the rapid antigen test was insensitive to samples with a ct higher than 30. The calculated specificity of the assay for asymptomatic patients remained at 99%.

## 4. Discussion

Overall results from the rapid antigen and PCR tests were generally concordant for these two large groups of individuals on the university campus, regardless of symptomology, when PCR ct were 30 cycles or less. Of note, asymptomatic subjects had higher underlying PCR cycle thresholds regardless of whether they were antigen-positive or negative, which has also been reported in a previous study [8]. Possibly the most significant observation from the study was that subjects missed with the rapid Ag test were generally only those with a PCR cycle threshold higher than 30 cycles, and possibly not infectious and/or less sick [9]. A ct of 30 cycles was chosen as the cut-off threshold as it reflects a potential viral load of 1000 virus particles, the minimum level thought to be needed for infectivity [10]. Overall, our findings based on validation samples as well as the analyses of symptomatic and asymptomatic samples indicated a sensitivity between 86 and 92% and a specificity of 97–99% for the rapid antigen test, for those individuals that are both infected and contagious with cycle thresholds under 30. That is, in symptomatic individuals the Ag test is very sensitive, while in asymptomatic individuals the Ag test detected less than half of those who had been infected with SARS-CoV-2 *but identified 86% of those who were contagious and likely to spread the virus*. In addition, based on the findings, a slightly higher false-negative rate for asymptomatic persons with the rapid antigen test could be overcome if one were to perform antigen testing a minimum of three times a week. Repeated testing could reduce to less than one percent the chance that any individual who is truly positive is missed by the antigen test [11].

Interestingly, more than half of all subjects who presented at Campus Health with COVID-19 symptoms during the study period did not test positive for SARS-CoV-2 by either a rapid antigen or PCR test, a noteworthy and cautionary finding in and of itself. This finding also helps to explain the lower-than-expected positivity rates in the symptomatic population. Our observations seem to indicate that compared with the rapid antigen test the PCR test detects viral loads that may be 200-fold lower, which may not be biologically relevant in terms of infectivity/contagiousness [9,10]. Recent evidence suggests that a PCR ct of 30 is the threshold for SARS-CoV-2 infectivity/contagiousness [10,12,13]. In fact, the rapid Ag test and the PCR test were generally concordant in their findings, although PCR testing may be overly sensitive (and the cut-off for PCR positivity may need to be lower than 40 cycles). The rapid Ag test may be best for determining if subjects are contagious and most likely to spread the virus, while the PCR test may be a better indicator for general infection and prevalence of the virus in the general population. Overall, the rapid antigen test seems to be sufficient and more than adequate for rapid screening of both asymptomatic and symptomatic subjects. Importantly, the rapid antigen test can easily be scaled up to test 2000–3000 subjects per day without a considerable increase in staff and equipment, making it amenable to high-throughput, large scale screening efforts (based on our own experience at the University of Arizona where we have now administered and performed more than 200,000 such rapid Ag tests). It also can be readily used for the daily screening of select populations with only a small decrease in sensitivity as compared to PCR testing. The decreased sensitivity can be compensated for with the increased test frequency of those individuals. However, it must be remembered that both the rapid Ag test and the PCR test are point-in-time tests, with no guarantee that one does not become infected moments to hours to days after sample collection, emphasizing the need for face coverings and social distancing to finally stem the pandemic until a vaccine becomes widely available.

## Figures and Tables

**Figure 1 biomedicines-09-00539-f001:**
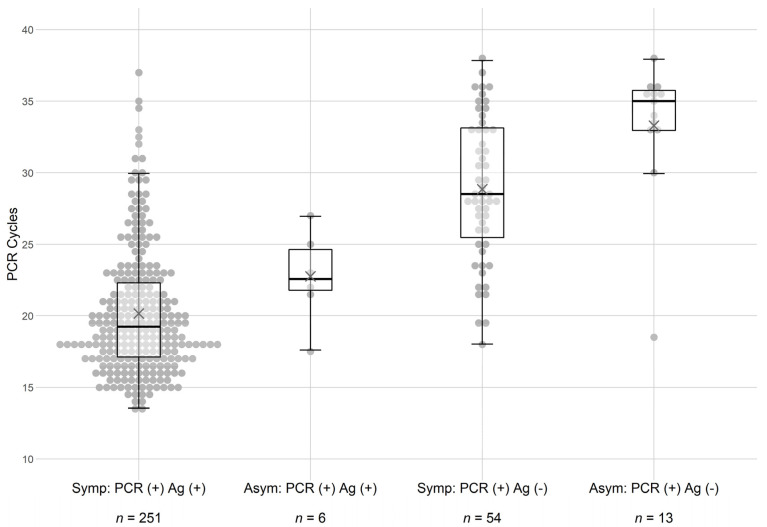
Comparison of PCR cycle thresholds in symptomatic and asymptomatic patients. Subjects were tested simultaneously by Ag and PCR testing for SARS-CoV-2. Patients were broadly grouped into symptomatic patients presenting at campus health and asymptomatic subjects on the university campus. Subgroup results were analyzed based on the combination of results for each test. Violin plots of the data are shown. Each circle represents an independent measurement for a subject.

**Table 1 biomedicines-09-00539-t001:** Validation of the SARS-CoV-2 rapid antigen test.

ID#	Type of Sample	Expected Result	Ag Test Result
1	SPK	POS	POS
2	SPK	NEG	NEG
3	SPK	POS	POS
4	SPK	POS	POS
5	SPK	POS	POS
6	SPK	POS	POS
7	SPK	NEG	NEG
8	SPK	POS	POS
9	SPK	POS	POS
10	SPK	POS	POS
11	SPK	POS	POS
12	SPK	NEG	NEG
13	SPK	NEG	NEG
14	SPK	POS	POS
15	SPK	POS	POS
16	SPK	POS	POS
17	SPK	POS	POS
18	SPK	NEG	NEG
19	SPK	POS	POS
20	SPK	POS	POS
**ID#**	**Type of Sample**	**Ag Test**	**RT-PCR (ct)**
BF00043	SYMP	POS	POS (ct 23)
BF00044	SYMP	POS	POS (ct 14)
BF00081	SYMP	POS	POS (na)
BF00094	SYMP	POS	POS (na)
BF00106	SYMP	POS	POS (na)
BF00112	SYMP	POS	POS (na)
22	ASYM	POS	POS (ct 27)
34	ASYM	POS	POS (ct 22)
113	ASYM	POS	POS (ct 25)
127	ASYM	POS	POS (ct 17)
156	ASYM	POS	POS (ct 22)
**112**	**ASYM**	**NEG**	**POS (ct 36)**
**139**	**ASYM**	**NEG**	**POS (ct 31)**
24524	ASYM	POS	POS (ct 23)
**21448**	**ASYM**	**NEG**	**POS (ct 35)**

A mixed set of possible positive and negative nasal swabs was analyzed as part of the validation of the rapid antigen test. Nasal swabs were collected as per the manufacturer’s instructions from “spiked” samples (SPK), symptomatic critically ill patients hospitalized in an intensive care unit (SYMP), and asymptomatic (ASYM) subjects from the campus community. Additional subjects that tested negative by both Ag and PCR tests during the validation studies are not shown. ct: PCR cycle threshold. POS: positive result/detected. NEG: negative result/not detected. ID#: unique sample identification number. na: ct not available due to HIPAA restrictions. Discordant results are shown in bold type.

**Table 2 biomedicines-09-00539-t002:** Longitudinal study of selected asymptomatic subjects.

ID#	Ag Test	RT-PCR (Cycle Threshold)
22	POS	POS (ct27)
	NEG	POS (ct33)
	NEG	POS (ct34)
113	POS	POS (ct25)
	NEG	POS (ct35)
127	POS	POS (ct17)
	NEG	POS (ct36)
	NEG	POS (ct35)
112	NEG	POS (ct36)
	NEG	POS (ct36)
139	NEG	POS (ct31)
	NEG	POS (ct36)

During the validation studies, several subjects were tested weekly for the virus over a period of 2–3 consecutive weeks, simultaneously by both antigen and PCR testing, with results shown above (tests conducted 1 week apart). At this time there were also additional samples that tested negative by both antigen and PCR tests that are not shown for the sake of clarity.

**Table 3 biomedicines-09-00539-t003:** Relationship between PCR cycle threshold and rapid antigen test reactivity.

Symptomatic Subjects		
PCR RESULTS		
**POS (average ct = 20)**		NEG
Ag Pos	*n* = 251	*n* = 7
**Ct < 30, *n* = 243, X = 20**		
**Ct > 30, *n* = 8, X = 33**		
Ag Neg	*n* = 54	*n* = 573
**Ct < 30, *n* = 34, X = 26**		
**Ct > 30, *n* = 20, X = 34**		
**Asymptomatic Subjects**		
**PCR RESULTS**		
**POS (average ct = 23)**		NEG
**Ct < 30, *n* = 6, X = 23**		
**Ct > 30, *n* = 0**		
Ag Neg	*n* = 13	*n* = 1525
**Ct < 30, *n* = 1, X = 18**		
**Ct > 30, *n* = 12, X = 35**		

Data presented in Table 3 are stratified according to PCR cycle thresholds (ct). Subjects that tested PCR-positive and Ag-negative (in bold) were divided into two groups: <30 ct and >30 ct and mean (X) ct values were calculated.

## Data Availability

Not applicable.

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
