# Peer review of "SARS-CoV-2 Rapid Antigen Testing of Symptomatic and Asymptomatic Individuals on the University of Arizona Campus"

_biomedicines, 2021, doi:10.3390/biomedicines9050539_

Round 1

Reviewer 1 Report

The paper summarizes the results of a field validation of a rapid  antigen test for Sars Cov2  detection (Quidel Rapid Ag test). Test sensitivity and specificity were calculated by testing a panel of asymptomatic an symptomatic subjects which were also screened for Sars-Cov2 presence by PCR.

The Ag test sensitivity and specificity were confronted with the results of the PCR assay, whose positive responses were stratified according to the positive cycle thresholds (Ct). In general, the Ag test was poorly sensitive on patients resulting PCR positive with high Cts (Ct>30).

The authors conclude that the Ag test is sensitive and specific on symptomatic subjects. On non symptomatic patients, it is capable to detect the virus only when its load is high enough for the subject to considered contagious.

The topic of the paper is of high interest at present times and the results are relevant for the general audience.

However, I found several minor flaws which should be fixed before publishing.

  1. Line 153: what exactly do you mean for 'exposed tothe virus? In table 3, 885 is the sum for the symptomatic. please  use  more clear and univocal definitions.
  2. lines 159-161: the sentence is unclear, it should be edited for clarity
  3. lines 163 – 164: Please check for correctness of sensitivity/specificity, NPV and PPV values on symptomatic subjects. Numbers do not seem correct. Eg. sensitivity on symptomatic should be 100-(54+251)*100 =82,3%; specificity on symptomatic should be calculated on PCR negatives, not on total samples = 100-(7/(7+573)x 100 = 98.2%;  
  4. For clarity purposes, I also suggest to add a final table summarizing sensitivity and specificity values on both symptomatic and NON symptomatic subjects, with values calculated a) independently of Ct, b) stratified for Ct>30 and c) stratified for Ct<30.   These are the numbers discussed on page 4. A table would provide more complete elaborated data and make the paper more easy to read.
  5. Ag test positivity/negativity response is based on the ratio of relative fluorescence units to background in a lateral flow test assay (the S/CO ratio) which, for positive subjects must be above 1 and with overall values between 1 to 600 depending on level of viral infection. Was a correlation found between the ratios measured and the PCR Ct?  Could you provide some information on this at least in a supplementary file?

Minor formalities

  1. Many abbreviations lack description (CLIA, CAP, CDC, ICU);
  2. Temperature (line 104) should be in Celsius;
  3. Table 2: indicate the interval time between the three tests.

Author Response

  1. Line 153: what exactly do you mean for 'exposed tothe virus? In table 3, 885 is the sum for the symptomatic. please  use  more clear and univocal definitions.  We have indicated in the text that exposed to the virus means in close contact with someone who tested positive for the virus.  Thank you for the comment about Table 3.
  2. lines 159-161: the sentence is unclear, it should be edited for clarity.  Thank you, we have added additional text and hopefully the intention is now clear.
  3. lines 163 – 164: Please check for correctness of sensitivity/specificity, NPV and PPV values on symptomatic subjects. Numbers do not seem correct. Eg. sensitivity on symptomatic should be 100-(54+251)*100 =82,3%; specificity on symptomatic should be calculated on PCR negatives, not on total samples = 100-(7/(7+573)x 100 = 98.2%;  Thank you for pointing out this error.  It has now been corrected.
  4. For clarity purposes, I also suggest to add a final table summarizing sensitivity and specificity values on both symptomatic and NON symptomatic subjects, with values calculated a) independently of Ct, b) stratified for Ct>30 and c) stratified for Ct<30.   These are the numbers discussed on page 4. A table would provide more complete elaborated data and make the paper more easy to read.  We are unsure as to what the reviewer would want to have inserted, and we are unsure that it is needed.
  5. Ag test positivity/negativity response is based on the ratio of relative fluorescence units to background in a lateral flow test assay (the S/CO ratio) which, for positive subjects must be above 1 and with overall values between 1 to 600 depending on level of viral infection. Was a correlation found between the ratios measured and the PCR Ct?  Could you provide some information on this at least in a supplementary file?  We did not investigate any correlation between ct values and viral load.  However, Quidel has investigated this relationship and has reported that higher S/CO values are associated with higher viral loads.  As we did not perform those experiments, we  really cannot insert such a table into the paper.

Minor formalities

  1. Many abbreviations lack description (CLIA, CAP, CDC, ICU); We have inserted a list of abbreviations as requested.
  2. Temperature (line 104) should be in Celsius; we have now insert C as well as F for temperatures.
  3. Table 2: indicate the interval time between the three tests. we have now indicated that the time between tests was 1 week.

Reviewer 2 Report

Biomedicines-1200830-peer-review 

Title: SARS-CoV-2 Rapid Antigen Testing of Symptomatic and 2 Asymptomatic Individuals on the University of Arizona Cam- 3 pus 

  1. The statistical information regarding the Sars-Cov-2 should be updated. The info is based on December 2020.
  2. It would be easier for the readers if authors can provide more information regarding the Quidel Sofia 2 test, for instance S/CO ratio could be explained and not explained abbreviations should be avoided throughout the manuscript.
  3. Could authors be more specific about the timing (“later times”) mentioned in lines 147-148?
  4. It would be helpful if authors compare Sofia 2 (sensitivity and specificity) with other commercially available rapid sars-cov-2 antigen tests got EUA.
  5. Authors suggested the use of rapid antigen test three times a week minimum to detect asymptomatic infections. Are there any experimental finding (i.e time course analysis with the kit) supporting this suggestion?
  6. Reference list should be extended with reviews and articles include comparative analysis of molecular and immunological methods for Covid-19. 

Author Response

  1. The statistical information regarding the Sars-Cov-2 should be updated. The info is based on December 2020.  We have updated the figures to February 2021 as requested.
  2. It would be easier for the readers if authors can provide more information regarding the Quidel Sofia 2 test, for instance S/CO ratio could be explained and not explained abbreviations should be avoided throughout the manuscript.  We have now provided more detail about the S/CO as requested and inserted a list of abbreviations used as suggested.
  3. Could authors be more specific about the timing (“later times”) mentioned in lines 147-148?  We have added more detail to this statement to reflect that later times means when ct values were tested to be higher.
  4. It would be helpful if authors compare Sofia 2 (sensitivity and specificity) with other commercially available rapid sars-cov-2 antigen tests got EUA.  We do not find this action to be really possible as we have not performed a detailed in-house validation of other rapid antigen tests and have found that depending on manufacturer's claims to be unreliable.
  5. Authors suggested the use of rapid antigen test three times a week minimum to detect asymptomatic infections. Are there any experimental finding (i.e time course analysis with the kit) supporting this suggestion?  Our suggestion is based on work by Mina et al, as well as what is known about time from initial infection to being contagious (2-3 days) and probable asynchonous entry into a setting where infections could spread.
  6. Reference list should be extended with reviews and articles include comparative analysis of molecular and immunological methods for Covid-19. Our manuscript was not concerned about differences between molecular and immunological assays for COVID-19, beyond the extent it was compared in the enclosed paper for the Quidel assay versus the CDC PCR assay.  It does not seem necessary to insert a review paper aspect into this succinct experimental report.

Round 2

Reviewer 2 Report

The paper can be published in its current form.